# Possible Role of Insulin-Degrading Enzyme in the Physiopathology of Retinitis Pigmentosa

**DOI:** 10.3390/cells11101621

**Published:** 2022-05-12

**Authors:** Alonso Sánchez-Cruz, María D. Hernández-Fuentes, Cayetana Murillo-Gómez, Enrique J. de la Rosa, Catalina Hernández-Sánchez

**Affiliations:** 1Department of Molecular Biomedicine, Centro de Investigaciones Biológicas Margarita Salas (CSIC), 28040 Madrid, Spain; asanchezcruz@cib.csic.es (A.S.-C.); mariadhfuentes@cib.csic.es (M.D.H.-F.); cmurillo@cib.csic.es (C.M.-G.); ejdelarosa@cib.csic.es (E.J.d.l.R.); 2Centro de Investigación Biomédica en Red de Diabetes y Enfermedades Metabólicas Asociadas (CIBERDEM-ISCIII), 28034 Madrid, Spain

**Keywords:** insulin-degrading enzyme, retinitis pigmentosa, retina, neurodegeneration, *rd1*, *rd10*, *P23H*, preimplantation factor

## Abstract

Insulin-degrading enzyme (IDE) was named after its role as a proteolytic enzyme of insulin. However, recent findings suggest that IDE is a widely expressed, multitask protein, with both proteolytic and non-proteolytic functions. Here, we characterize the expression of IDE in the mammalian retina in both physiological and pathological conditions. We found that IDE was enriched in cone inner segments. IDE levels were downregulated in the dystrophic retina of several mouse models of retinitis pigmentosa carrying distinct mutations. In *rd10* mice, a commonly studied mouse model of retinitis pigmentosa, treatment with an IDE activator (a synthetic peptide analog of preimplantation factor) delayed loss of visual function and preserved photoreceptor cells. Together, these results point to potential novel roles for IDE in retinal physiology and disease, further extending the list of diverse functions attributed to this enzyme.

## 1. Introduction

Insulin-degrading enzyme (IDE) is a ubiquitously expressed zinc metalloendopeptidase [1]. It was initially identified as an insulin-clearing enzyme (the role for which it was named) [2]. However, IDE is currently considered a multitask enzyme with recognized proteolytic and non-proteolytic functions [3]. As a protease, IDE digests a variety of targets based on substrate tertiary structure constraints rather than amino acid sequence specificity. In addition to insulin, IDE has been shown to degrade hormones, cytokines, growth factors, and amyloid peptides [3]. Moreover, IDE interacts in a non-proteolytic manner with diverse intracellular molecules and proteins, although the outcome of most of those interactions is unknown [3]. The broad proteolytic and non-proteolytic substrate spectrum of IDE suggests a role in the regulation of multiple cellular functions in addition to insulin clearance. However, there is little, if any, definitive data on the pathophysiological roles of IDE [4].

IDE is prominently expressed in the brain [1], and in recent years has attracted considerable attention as a potential therapeutic target for Alzheimer’s disease, owing to its role in Aβ amyloid clearance [5,6]. The broad expression of IDE in the brain suggests that it may participate in several other functions. 

The retina is a highly specialized part of the central nervous system (CNS) that is responsible for converting light stimuli into chemo-electrical signals, which are then sent to the brain for further processing. As part of the CNS, the retina shares multiple pathophysiological features with the brain [7]. As in the brain, neurodegenerative processes in the retina ultimately cause irreversible loss of neurons, leading to retinal dysfunction and major disabilities. Remarkably, retinal structure and function can be assessed by non-invasive techniques, providing a useful experimental system with which to characterize physiopathological conditions that affect the CNS, as well as potential therapies.

Retinitis pigmentosa (RP) is a group of retinal genetic dystrophies responsible for the majority of hereditary blindness cases. RP is considered a rare disease, with a prevalence of 1 per 3000–4000 [8], and encompasses a range of genetically heterogeneous disorders caused by more than 3000 different mutations in over 60 genes (https://sph.uth.edu/retnet/sum-dis.htm (accessed on 4 March 2022)). Despite its genetic heterogeneity, most forms of RP involve primary dysfunction and death of rod photoreceptors, resulting in night blindness, followed by secondary loss of cones that leads to constriction of the visual field and eventually to blindness. Currently, RP is neither preventable nor curable. Although gene therapy holds the greatest potential for definitive treatment, gene-independent therapies constitute a valuable initial alternative given the complex genetic etiology of RP. 

We have previously shown that proinsulin, the precursor of insulin, acts as a neuroprotective factor in the dystrophic retina and ameliorates RP progression [9,10,11]. In exploratory studies, we found that proinsulin increases IDE levels in the retina of the *rd10* mice, a model of RP. This prompted us to study the potential role of IDE in the physiopathology of the retina.

In the present study, we investigated the retinal expression of IDE in physiological conditions (wild-type (WT) mice) and in the context of RP in the *rd1*, *rd10*, and *P23H/P23H* mouse models of this disease, which carry distinct mutations that cause the disease in humans and reproduce the clinical hallmarks of RP. We show that IDE is enriched in the cone inner segments (ISs). The gene expression and protein levels of IDE are reduced in both *rd10* and *P23H/P23H* mouse retinas with respect to age-matched WT retinas. Moreover, treatment with a synthetic peptide analog of the preimplantation factor (sPIF), an activator of IDE [12], results in better preservation of retinal structure and function in the *rd10* mice. Overall, these results reveal a novel potential role of IDE in the physiopathology of the retina.

## 2. Materials and Methods

### 2.1. Animals

*rd1*, *rd10*, *P23H/P23H*, and WT control mice were obtained from The Jackson Laboratory (Bar Harbor, ME, USA). The *rd1* (*Pde6b^rd1/rd1^*) and *rd10* (*Pde6b^rd10/rd10^*) mice carry a recessive homozygous spontaneous nonsense and missense mutation, respectively, in the rod-specific phosphodiesterase 6b gene [13,14]. The *P23H/P23H* (*Rho^P23H/P23H^*) mice were generated by a knock-in strategy and carry a dominant mutation in which the proline at position 23 of the rhodopsin gene has been replaced with a histidine [15]. The *rd1* mice present the earliest disease onset, and most rods are lost by postnatal day (P) 15. Both the *rd10* and *P23H/P23H* models also display rapid disease progression, leading to blindness early in life. In the *rd10* mice, most rod photoreceptors are lost between P18 and P30. In the *P23H/P23H* mice, most rods die between P14 and P21. All animals were bred on a C57BL/6J background, and were housed and handled in accordance with the ARVO statement for the Use of Animals in Ophthalmic and Vision Research and the guidelines of the European Union and the local ethics committees of the CSIC and the Comunidad de Madrid (Ref: PROEX 287/19, 17 February 2020; PROEX 272.8/21, 1 October 2021). Mice were bred and housed at the CIB Margarita Salas core facilities on a 12/12 h light–dark cycle. Light intensity was maintained at 3–5 lux.

### 2.2. Electroretinography (ERG) Recordings

Electroretinographic responses were recorded using a device designed by Dr. P. de la Villa (Universidad de Alcalá, Madrid, Spain). ERG signals were amplified and band filtered between 0.3 Hz and 1000 Hz using a PowerLab T15 acquisition data card (AD Instruments Ltd., Oxfordshire, UK). Mice were maintained in darkness overnight. The next day, animals were anesthetized in scotopic conditions with ketamine (50 mg/kg; Ketolar, Pfizer, New York, NY, USA) and medetomidine (0.3 mg/kg; Domtor, Orion Corporation, Espoo, Finland), and their pupils dilated with a drop of tropicamide (Alcon, Fort Worth, TX, USA). Next, the ground electrode was located parallel to the tail of the animal, and the reference electrode was placed in the mouth. Methocel (Colorcon, Harleysville, PA, USA) was applied to the cornea to avoid drying, and the corneal electrode was placed in contact with the Methocel. ERG recordings were first obtained in scotopic conditions with increasing light stimuli (0.1, 1, 10, and 50 cd·s/m^2^). After light adaptation at 30–50 cd/m^2^ (photopic conditions), ERG response was measured at increasing light stimuli (1, 10, and 50 cd·s/m^2^). After ERG recording, sedation was interrupted with atipamezol (1 mg/kg; Antisedan, Orion Corporation, Espoo, Finland). All measurements were performed by an observer blind to the experimental condition. Wave amplitude was analyzed using Labchart 7.0 software (AD Instruments, Oxford, UK).

Mice were euthanized the day after the ERG; one eye was processed for histological analysis and the other for either RNA isolation and quantitative PCR (qPCR) or protein extraction, as described below.

### 2.3. RNA Isolation and Quantitative PCR

Total RNA was isolated from tissues using Trizol reagent (ThermoFisher Scientific, Boston, MA, USA). Before reverse transcription (RT), potentially contaminating DNA was eliminated with DNAse I (ThermoFisher Scientific). RT was performed with 1 μg of RNA and with the Superscript III Kit and random primers (all from ThermoFisher Scientific). qPCR was performed with the ABI Prism 7900HT Sequence Detection System using TaqMan Universal PCR Master Mix, no-AmpEthrase UNG, and Taqman assays (listed in Table 1) for detection (all from ThermoFisher). Relative changes in gene expression were calculated using the ΔCt method, normalizing to expression levels of the *Tbp* (TATA-binding protein) gene.

### 2.4. Histological Analysis of Retinal Sections

Animals were euthanized, and their eyes were enucleated and fixed for 50 min in freshly prepared 4% paraformaldehyde in Sörensen’s phosphate buffer (SPB) (0.1 M, pH 7.4) and then cryoprotected by incubation in increasing concentrations of sucrose (final concentration, 50% in SPB), all at room temperature. Eyes were then embedded in Tissue-Tek OCT (Sakura Finetec, Torrance, CA, USA) and snap-frozen in isopentane on dry ice. Equatorial sections (12 µm) were cut on a cryostat and mounted on Superfrost Plus slides (ThermoFisher Scientific), dried at room temperature, and stored at −20 °C until the day of the assay. Before performing further analyses, slides were dried at room temperature. After rinsing in PBS and permeation with 0.2% (*w*/*v*) Triton X-100 in PBS, sections were incubated with blocking buffer (5% (*v*/*v*) normal goat serum, 1% (*w*/*v*) Tween-20, 1 M glycine in PBS) for 1h at room temperature and then incubated overnight at 4 °C with primary antibodies (Table 2) diluted in blocking buffer. Sections incubated with normal rabbit serum were used as specificity control for IDE antibody. Sections incubated in the absence of primary antibody were used as specificity control for the rest of antibodies. After rinsing in PBS and incubation with the appropriate secondary antibodies (Table 2) for 2 h at room temperature, sections were stained with DAPI (4′,6-diamidino-2-phenylindole; Sigma-Aldrich Corp., St. Louis, MO, USA) and cover-slipped with Fluoromount-G (ThermoFisher Scientific). 

Sections were analyzed using a laser confocal microscope (TCS SP5 and TCS SP8; Leica Microsystems, Wetzlar, Germany). In all cases, retinal sections to be compared were stained and imaged under identical conditions. For the quantification of IDE intensity in the IS of cones, we used the “freehand line” tool in Fiji software (https://imagej.net/software/fiji/, accessed on 4 March 2022) to select the whole length of the cone IS. Next, we analyzed the plot profile to quantify IDE intensity along the cone IS. Finally, we measured the area under the curve of the previous plot, which allowed us to quantify IDE expression in each cone IS. This value was recorded for a total of 20 cone ISs per mouse from 4 different images of the central retina. 

To assess preservation of the photoreceptor layer, we compared the thickness of the outer nuclear layer (ONL), which primarily contains photoreceptors, with that of the corresponding inner nuclear layer (INL), which contains bipolar, horizontal, and amacrine neurons and Müller glial cell bodies. ONL thickness was normalized to that of the INL, which was not affected by degeneration at this stage [15,16], in order to correct for possible deviations in the sectioning plane. Two or three sections per retina were analyzed; for each section, six areas in a nasotemporal sequence were photographed [17]. ONL and INL thickness were measured in three random positions for each image. The length of rod and cone outer segments (OSs) was evaluated by measuring the length of rhodopsin, L/M cone, and S-cone staining. In each image, 3 measurements were recorded at random positions to obtain an average value per retinal zone per section. Measurements were acquired using the “freehand line” and “measure” tools in Fiji software.

### 2.5. Immunoblots

Protein extraction was carried out by sonicating individual retinas in RIPA lysis buffer (containing 2 mM Na_3_VO_4_, 10 mM NaF, and 4 mM Na_4_P_2_O_7_) and chilling the resulting solution on ice for 30 min. Protein (25 μg) from each sample was fractionated by electrophoresis on precast 10–12% (*w*/*v*) SDS-polyacrylamide gels (Criterion TGX, Bio-Rad, Munich, Germany), after which proteins were transferred to PVDF membranes using the Trans-Blot Turbo system (Bio-Rad). Blots were incubated overnight at 4 °C with primary antibodies (Table 2) diluted in TBS (Tris-buffered saline) 1% (*w*/*v*) Triton-X100, followed by incubation with the appropriate peroxidase-conjugated secondary antibody (Table 2). Proteins were visualized using the Pierce ECL Western Blotting Substrate (ThermoFisher Scientific, Waltham, MA, USA), and quantified using ChemiDoc Touch Imaging System (Bio-Rad). In all cases, the retinal protein extracts to be compared were run in the same gel and blotted and visualized simultaneously. Duplicates for each biological replicate were run and blotted, and the mean of the quantification of the duplicates was plotted for each biological replicate. 

### 2.6. Analysis of IDE Activity in Retinal Extracts

Analysis of IDE activity was performed by a fluorometric assay with SensoLyte 520 IDE activity assay kit (AS-72231, Anaspec, Fremont, CA, USA). Protein extraction was carried out by sonicating individual retinas in the commercial assay buffer. The activity of IDE was assayed following the manufacturer’s instructions, and fluorescence was detected with Varioskan LUX (ThermoFisher Scientific). IDE activity was normalized to total protein in the sample as quantified with Pierce BCA assay kit (ThermoFisher Scientific). IDE activity in an individual mouse corresponds to the mean value of both retinas.

### 2.7. Preimplantation Factor Treatment

Synthetic preimplantation factor (MVRIKPGSANKPSDD) was synthesized in the Protein Chemistry facility in the CIB Margarita Salas by solid-phase peptide synthesis (Peptide Synthesizer, AAPPtec, Focus XC; Louisville, KY, USA) employing Fmoc (9-fluorenylmethoxycarbonyl) chemistry. Final purification was carried out by reversed-phase HPLC, and identity was verified by linear ion trap LC/MS with electrospray ionization and amino acid analysis at >95% purity. 

Mice received daily intraperitoneal injections of vehicle (saline) or sPIF (1 mg/kg) for the indicated period of time.

### 2.8. Statistical Analysis

Statistical analyses were performed with GraphPad Prism software 8.0 (GraphPad Software Inc., La Jolla, CA, USA). For two-group comparisons, normality was first assessed using the Shapiro–Wilk normality test. The 2-tailed unpaired Student’s *t*-test and Mann–Whitney U-test were used to analyze normally and non-normally distributed data, respectively. Analysis of more than two datasets was performed using a one-way ANOVA. In these cases, Dunnett’s multiple comparison test was used to compare the values obtained for different experimental conditions with those of a control condition. Sidak’s post-test was used for multiple comparisons between preselected conditions. Comparisons of two variables were performed using a 2-way ANOVA, followed by Sidak’s multiple comparison post hoc test in cases in which a significant interaction between both variables was found. In all cases, statistical significance was established at *p* < 0.05.

## 3. Results

### 3.1. IDE Expression and Distribution in the WT Retina

We first evaluated physiological retinal *Ide* expression in the context of tissues known to express high *Ide* levels, specifically the brain and liver [1]. RT-qPCR analysis of adult mouse tissue confirmed that the highest *Ide* RNA levels were in the liver (Figure 1). In all CNS regions tested (telencephalon, cerebellum, olfactory bulb, and retina), *Ide* RNA levels were high, albeit lower than in the liver (Figure 1). 

Analysis of the distribution of retinal IDE expression by immunofluorescence revealed scattered intense staining located in the photoreceptor IS (Figure 2A). Colabeling of IDE and PNA, a marker of cones, revealed that this intense staining corresponded to the cone IS. Conversely, we observed no colocalization of IDE with rhodopsin or S-opsin (Figure 2B–D), which label the outer segments of rods and cones, respectively, thus confirming expression of IDE in the IS. In line with this observation, data from the Human Proteome Map portal (www.proteinatlas.org (accessed on 4 March 2022)) showed that *Ide* transcripts in the human retina are particularly abundant in cones, followed by rods (Figure 2E) [18]. 

### 3.2. Comparative IDE Expression in WT and RP Retinas

Next, we sought to characterize the expression of *Ide* in the context of the dystrophic retina. We first analyzed *Ide* RNA levels in retinas from three RP mouse models carrying distinct mutations: *rd1* (*Pde6b^rd1/rd1^*) and *rd10* (*Pde6b^rd10/rd10^*) mice, which carry nonsense and missense mutations in the *Pde6b* gene, respectively, and the *P23H/P23H* (*Rho^P23H/P23H^*) mice, which harbor a Pro23His replacement in the rhodopsin gene. *Ide* gene expression was significantly reduced in the degenerating retinas of all three RP models relative to those of age-matched WT controls (Figure 3A). Furthermore, Western blot analysis of retinal protein extracts from *P23H/P23H* and *rd10* mice indicated lower retinal IDE protein levels in both models, in line with *Ide* gene expression findings (Figure 3B–E). These results are in agreement with the reduced IDE activity in *rd10* versus WT retinas (Figure 3F). 

Furthermore, analysis of IDE expression in the *rd10* and *P23H/P23H* retinas revealed a significant decrease in IDE levels in the cone IS (Figure 4). Taken together, these results demonstrate an association between RP and the downregulation of retinal IDE, independent of the causative mutation.

### 3.3. Treatment with sPIF Ameliorated RP Progression

Based on the aforementioned findings, we next investigated whether stimulation of IDE impacts the progression of RP. To this end, we treated *rd10* mice with sPIF, one of the very few activators of IDE [12]. We selected the *rd10* mice for this experiment as this model has a wider therapeutic window than the *P23H/P23H* mice (see Materials and Methods). *rd10* littermates received daily intraperitoneal injections of either vehicle or sPIF (1 mg/kg) from P15 (i.e., before the appearance of evident retinal degeneration) to P27 (after most rod loss occurs) [16]. Visual function was assessed by ERG at P28. sPIF treatment resulted in a modest but significant increase in *Ide* gene expression. This effect correlated with higher IDE levels in the cone IS in sPIF-treated retinas and with a trend toward increased IDE activity in sPIF-treated retinal extracts (Appendix A). Compared with vehicle-treated counterparts, sPIF-treated *rd10* mice showed better-defined and larger-amplitude ERG waves (Figure 5A). Waves corresponding to both rods and cones (b-mixed waves) and cones (b-photopic waves) were significantly higher in amplitude in sPIF-treated versus control *rd10* mice (Figure 5B). We observed a trend towards increased flicker amplitude, which also reflects cone response, although this effect did not reach statistical significance (Figure 5B).

One hallmark of the different forms of RP is the primary death of rods followed by the cones, in more advanced stages, which results in the progressive decrease in the thickness of the photoreceptor layer (ONL). Similarly, photoreceptors undergo major morphological changes, such as the shortening and disorganization of their OS. Therefore, to investigate the effect of sPIF treatment on retinal cytoarchitecture preservation, two histological parameters were evaluated: ONL thickness as a readout of photoreceptor survival and OS length as an indicator of photoreceptor structure maintenance. The histological evaluation of retinas at the end point of the study (P29, after the ERG) revealed a greater relative ONL thickness in sPIF- versus vehicle-treated retinas, despite the advanced degeneration at this timepoint (Figure 6). Moreover, specific immunostaining for rod and cone OS showed better preservation of photoreceptor structure in sPIF-treated retinas (Figure 6). While all rods express rhodopsin, in the mouse retina, cones express L/M-opsin, S-opsin, or both; thus, by analyzing L/M- and S-opsin, we covered the entire cone population. Rhodopsin immunostaining in rod OS confirmed partial preservation in sPIF-treated retinas. By contrast, rod OSs were barely present in vehicle-treated retinas. Similarly, L/M- and S-opsin immunostaining revealed longer cone OSs in sPIF- versus vehicle-treated *rd10* retinas (Figure 6).

In agreement with the better photoreceptor preservation observed on histology, RT-qPCR analysis revealed higher expression of photoreceptor genes in sPIF-treated *rd10* retinas (Appendix A). 

Overall, these results suggest that sPIF stimulation of IDE has a disease-modifying effect over the course of RP. 

## 4. Discussion

More than 70 years after the first description of insulin-degrading enzyme [2], further findings have considerably broadened the spectrum of potential functions of IDE [3]. Moreover, as provocatively suggested by González-Casimiro [3] and Leissring [4], insulin turnover may not be the main physiological role of IDE. We found that retinal IDE expression is enriched in the cone IS. Likewise, retinal single-cell transcriptomic analysis showed that the highest levels of *Ide* expression in the human retina are in cones, followed by rods (www.proteinatlas.org (accessed on 4 March 2022)) [18]. In fact, according to the information reported in the database, cones express the second highest levels of IDE of all cells in the human body.

Photoreceptors are highly specialized neurons that generate chemo-electrical signals in response to stimulation by light. They consist of four primary regions: the outer segment, inner segment, cell body, and synaptic terminal. Most of the cell’s housekeeping functions, including protein and lipid synthesis, occur in the IS, which contains the Golgi, ER, and ribosomes, all of which participate in the daily renewal of the OS, and abundant mitochondria to meet the high energy demands of the phototransduction process. Interestingly, a mitochondrial IDE isoform generated via an alternative translation initiation site has been described [20], although its function and significance are unclear. Whether the high IDE levels found in the IS are in part due to the contribution of this mitochondrial IDE isoform merits further investigation. The IS also contains proteins involved in the formation of the basal structure of the OS, machinery for sorting and trafficking proteins involved in OS renewal, and supramolecular complexes comprising scaffolding and/or adhesion molecules [21,22]. Intriguingly, some of the non-proteolytic functions proposed for IDE include protein trafficking and subcellular localization, and scaffolding activities [3]. Moreover, shortening of the OS is a common histological alteration observed in different forms of RP. Here, we describe a decrease in IDE levels in the IS accompanied by OS shortening in two genetically unrelated mouse models of RP. Further studies will be required to determine whether there is a causal relationship between these two observations, and to characterize the potential roles of IDE in OS renewal. 

One of our main research objectives was to discover novel therapeutic targets for RP, a disabling and incurable disease for the great majority of the patients. The decreased IDE levels observed in the dystrophic retina prompted us to investigate whether IDE activation could have a disease-modifying effect. We found that treatment of *rd10* mice with sPIF, an IDE activator, ameliorated vision loss and preserved photoreceptor number and structure, suggesting that IDE may constitute a novel therapeutic target for RP. PIF is a natural peptide with neuroprotective functions [23] that increases the expression of and activates IDE [12]. Here, we show that sPIF treatment resulted in an increase in IDE expression and a trend towards increased IDE activity. We have found that proinsulin, which exerts neuroprotective effects in the dystrophic retina [9,10,11], also increases *Ide* levels (data not shown). Although the present findings point to a novel role of IDE in the retinal pathophysiology, any beneficial effects of IDE on retinal neurodegeneration will need to be corroborated in further studies using bona fide, selective in vivo IDE activators. Kraupner et al. [24] very recently identified potential IDE activators by screening drug libraries. Confirmation of the identified IDE molecules as bona fide in vivo activators could provide much needed tools to better probe the physiopathological role of IDE in different tissues, including the retina.

## Figures and Tables

**Figure 1 cells-11-01621-f001:**
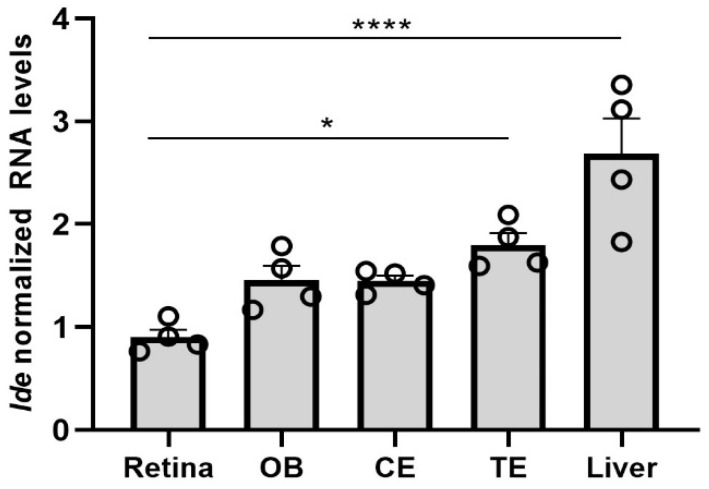
Expression of *Ide* in wild-type mouse tissues. Gene expression of *Ide* was analyzed by RT-qPCR in the indicated adult tissues. Dots represent individual animals, and bars represent the mean (+SEM) (*n* = 4 mice per group). Data were normalized to *Tbp* RNA levels. Gene expression was compared using a 1-way ANOVA, followed by Dunnett’s multiple comparison test. **** *p* < 0.0001; * *p* < 0.05. OB, olfactory bulb; CE, cerebellum; TE, telencephalon.

**Figure 2 cells-11-01621-f002:**
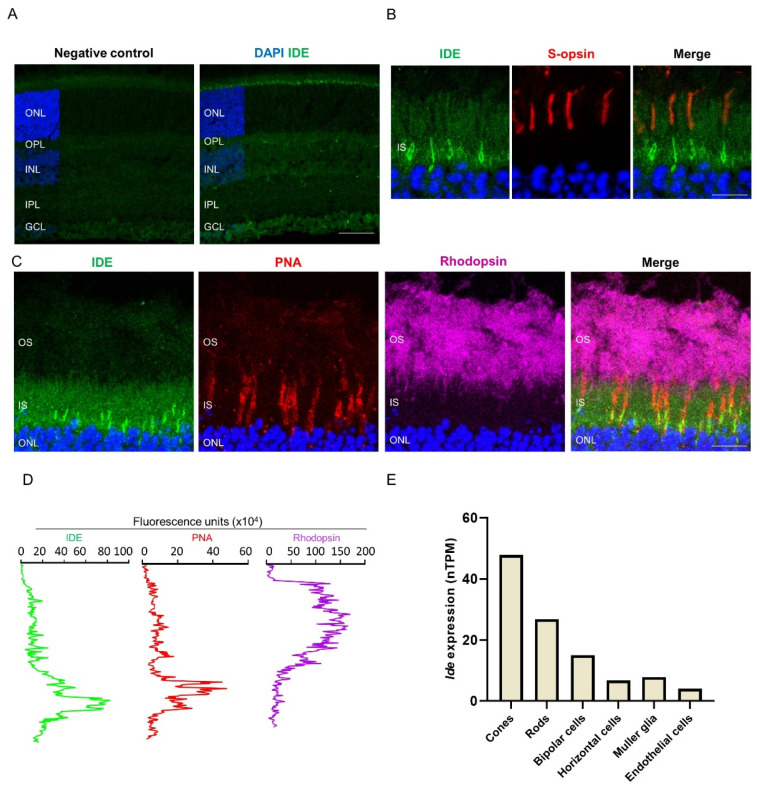
IDE distribution in the wild-type retina. (**A**) Retinal section at P25 immunostained for IDE (green). Nuclei were stained with DAPI (blue). In the negative control, the IDE primary antibody was substituted by normal rabbit serum. (**B**) Magnified image of photoreceptor segments showing double staining for IDE (green) and S-opsin (red). (**C**) Magnified image of photoreceptor segments showing triple staining for IDE (green), PNA (red), and rhodopsin (magenta). (**D**) Diagram shows fluorescence intensity of each fluorophore along the “Y” axis of panels shown in C. OS, outer segment; IS, inner segment; ONL, outer nuclear layer; OPL, outer plexiform layer; INL, inner nuclear layer; IPL, inner plexiform layer; GCL, ganglion cell layer. Scale bars: 60 µm (**A**), 9 µm (**B**), and 13 µm (**D**). (**E**) *Ide* expression in human individual retinal cell types. Data obtained by single-cell RNA sequencing and extracted from the Human Proteome Map portal (www.proteinatlas.org (accessed on 4 March 2022)). nTPM, normalized transcripts per million.

**Figure 3 cells-11-01621-f003:**
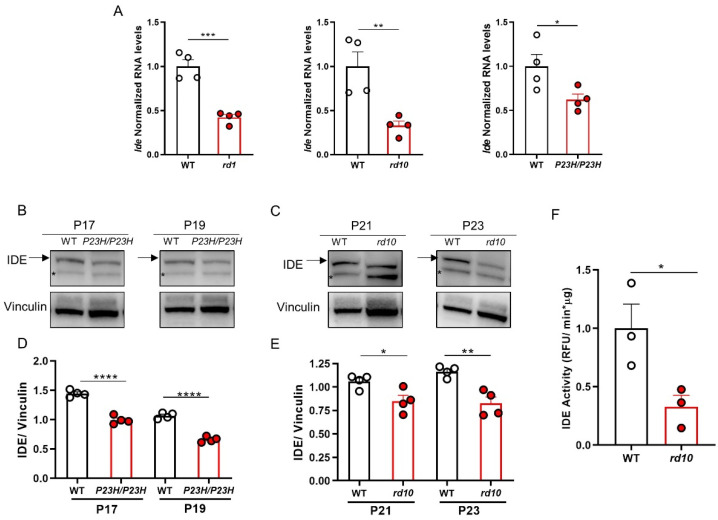
Comparison of retinal IDE levels in wild-type mice and mouse models of retinitis pigmentosa. (**A**) RT-qPCR of individual retinas from WT and *rd1* mice at P11, WT and *rd10* mice at P21, and WT and *P23H/P23H* mice at P14. *Ide* transcript levels were normalized to *Tbp* RNA levels and expressed relative to corresponding WT levels (assigned a value of 1). Results represent the mean + SEM. *n* = 4 individual retinas from 4 mice. *** *p* < 0.001; ** *p* < 0.01; * *p* < 0.05 (unpaired 2-tailed Student’s *t*-test). (**B**,**C**) Representative images showing Western blots of individual retinas from WT and *P23H/P23H* (**B**) or WT and *rd10* (**C**) mice at the indicated ages. Arrow indicates the 110 kDa band corresponding to IDE, and asterisk indicates a non-specific band, previously described by Fernández-Díaz et al. [19]. (**D**,**E**) Densitometric analysis of Western blot images. Levels of IDE were normalized to those of vinculin. Results represent the mean + SEM. *n* = 4 individual retinas from 4 mice, **** *p*< 0.0001; ** *p* < 0.01 (2-way ANOVA). (**F**) IDE activity as measured by fluorometric assay of WT and *rd1*0 retinas at P23. Results represent the mean + SEM. *n* = 3 mice. * *p* < 0.05 (unpaired 2-tailed Student’s *t*-test).

**Figure 4 cells-11-01621-f004:**
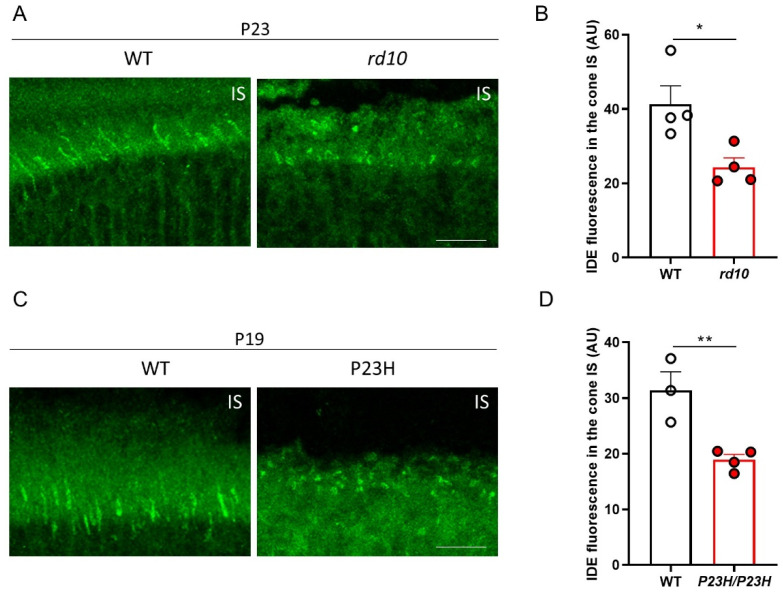
Comparison of IDE expression in the cone IS of wild-type mice and 2 mouse models of retinitis pigmentosa. (**A**,**C**) Images showing WT and *rd10* retinal sections at P23 (**A**) and WT and *P23H/P23H* retinal sections at P19 (**C**) immunostained for IDE (green). (**B**,**D**) Quantification of IDE fluorescence intensity in the cone IS in WT and *rd10* retinas at P23 (**B**) and WT and *P23H/P23H* retinas at P19 (**D**). *n* = 3–4 individual retinas from 3–4 mice. ** *p* < 0.01; * *p* < 0.05 (unpaired 2-tailed Student’s *t*-test). IS, inner segment; AU, arbitrary units. Scale bars: 13 µm.

**Figure 5 cells-11-01621-f005:**
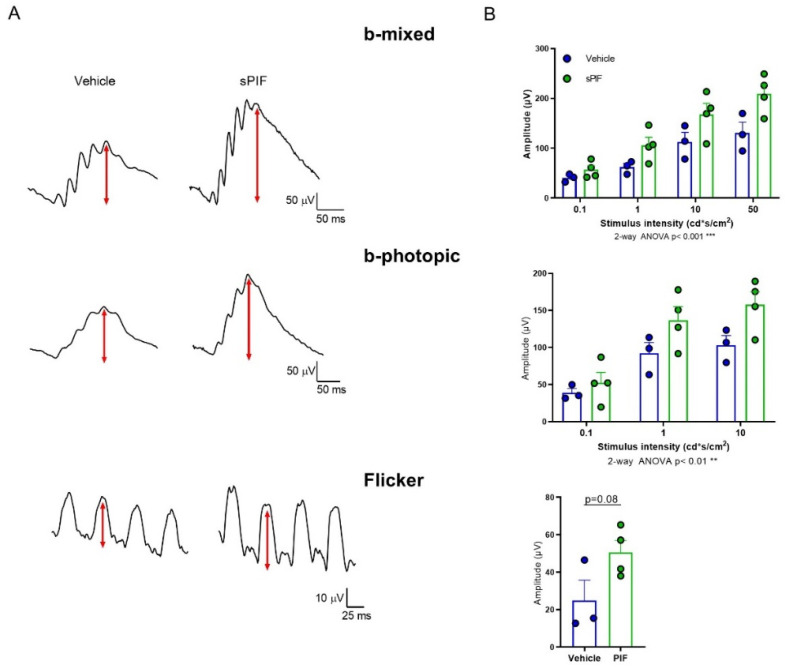
Effect of sPIF treatment on retinal response to light in *rd10* mice. Animals received daily intraperitoneal injections of vehicle or sPIF from P15 to P27. ERG recordings were performed at P28. (**A**) Representative ERG responses of vehicle- and sPIF-treated mice to a stimulus of 50 cd·s/m^2^. Scale bar value is indicated in the figure. (**B**) Graphs show mean ERG wave amplitudes, plotted as a function of light stimulus. Amplitudes of rod and cone mixed responses (b-mixed waves) to the indicated light intensities were recorded under scotopic conditions after overnight adaptation to darkness. Amplitudes of cone responses (b-photopic and flicker waves) to the indicated light intensities were recorded under photopic conditions after 5 min of light adaptation (30 cd·s/m^2^). Dots represent individual mice, and bars represent the mean (+SEM) for each group. *n* = 3–4 animals per group. *** *p*< 0.001; ** *p*< 0.01 (2-way ANOVA); *p* = 0.08 (unpaired 2-tailed Student’s *t*-test).

**Figure 6 cells-11-01621-f006:**
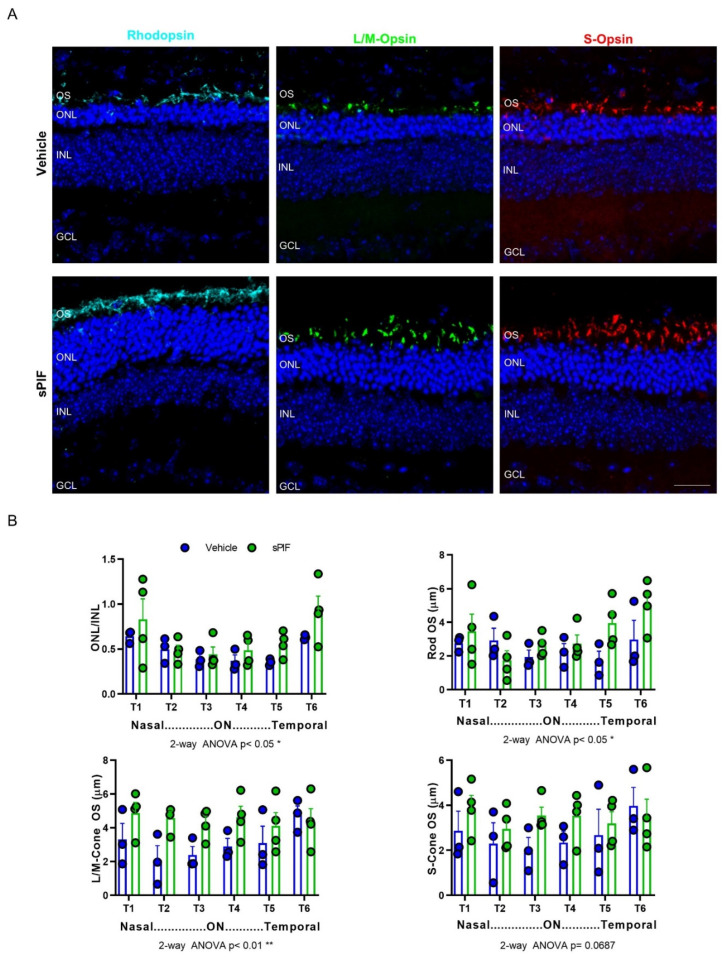
Effect of sPIF treatment on photoreceptor preservation in *rd10* mice. Animals received daily intraperitoneal injections of vehicle or sPIF from P15 to P27. Retinas were analyzed at P29, after ERG had been performed. (**A**) Representative images of central retinal areas in cryosections immunostained for rhodopsin (cyan), L/M-opsin (green), and S-opsin (red). Nuclei were stained with DAPI (blue). ONL, outer nuclear layer; INL, inner nuclear layer, GCL, ganglion cell layer. Scale bar, 22 μm. (**B**) ONL and INL thickness and the length of rod and cone outer segments were measured in equatorial sections corresponding to 6 regions of the retina, following a nasotemporal sequence (T1–T6). Plots show the mean + SEM. *n* = 3–4 mice, 2–3 sections per retina, 3 measurements per region and section. ON, optic nerve. ** *p* < 0.01; * *p* < 0.05 (2-way ANOVA).

**Table 1 cells-11-01621-t001:** TaqMan assays used in the experiments.

Gene	Probe
*Arr3* (cone arrestin)	Mm00504628_m1
*Ide* (Insulin-degrading enzyme)	Mm00473077_m1
*Opn1mw* (L/M-Opsin)	Mm00433560_m1
*Opn1sw* (S-Opsin)	Mm00432058_m1
*Rcvrn* (Recoverin)	Mm00501325_m1
*Rho* (Rhodopsin)	Mm01184405_m1
*Tbp* (TATA-binding protein)	Mm01277042_m1

**Table 2 cells-11-01621-t002:** Antibodies used in the experiments.

Antibody (Cell-Type Specificity)	Host	Dilution	Manufacturer	Catalog
IDE	Rabbit	IH, 1:200 WB, 1:1000	Millipore, Burlington, MA, USA	AB9210
L/M-opsin (Cones)	Rabbit	IH, 1:200	Millipore, Burlington, MA, USA	AB5405
PNA-Alexa 488 (Cones)	Peanut	IH, 1:250	ThermoFisher Scientific, Waltham, MA	L21409
Rhodopsin (Rods)	Mouse	IH, 1:500	Abcam, Cambridge, UK	AB3267
S-opsin (Cones)	Goat	IH, 1:200	Santa Cruz, Santa Cruz, CA, USA	SC14363
Vinculin	Mouse	WB, 1:1000	Merck, Darmstadt, Germany	V9131
Anti-Igs- Alexa 488-546-647	Goat	IH, 1:200–500	ThermoFisher Scientific, Waltham, MA	A-11001 A-11008 A-11004 A-11011 A-21235
Anti-Igs-Alexa 568	Donkey	IH, 1:200	ThermoFisher Scientific	A-11057
Anti-mouse immunoglobulins/HRP	Rabbit	WB, 1:5000–1:10,000	Dako, Santa Clara, CA, USA	P0161
Anti-rabbit immunoglobulins/HRP	Goat	WB, 1:5000–1:10,000	Dako, Santa Clara, CA, USA	P0448

## Data Availability

The datasets used and/or analyzed during the current study are available from the corresponding author on reasonable request.

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
