# Peer review of "Possible Role of Insulin-Degrading Enzyme in the Physiopathology of Retinitis Pigmentosa"

_cells, 2022, doi:10.3390/cells11101621_

Round 1

Reviewer 1 Report

accept in present form

Author Response

We thank the reviewer for accepting our manuscript for publication in Cells. 

Reviewer 2 Report

Review of the manuscript # cells-1649249 titled: Possible role of insulin-degrading enzyme in the physiopathology of retinitis pigmentosa.

In this manuscript, Sánchez-Cruz and colleagues explore the role of insulin degrading-enzyme (IDE) in the mammalian retina and its implication in retinitis pigmentosa (RP). The authors studied the expression and distribution of IDE in physiological and pathological conditions. For this reason, they used three RP mice models (rd1, rd10, P23H/P23H) that differ in the causative mutation. They found lower levels of IDE expression and protein in dystrophic retinas than in wild types (WT). Additionally, their results exhibit a decrease of IDE expression in the inner segment (IS) and the ganglion cell layer (GCL) of rd10 and P23H/P23H, whereas they previously showed that both areas present elevated IDE expression in WT. Then the authors analyzed whether IDE activation could reduce the pathological effects on the mammalian retina, using a synthetic preimplantation factor (sPIF), which has an activator effect over IDE expression. Interestingly, they found that sPIF treatment in rd10 mice improves the retinal response to light and promotes the preservation of photoreceptors.

Overall, the manuscript could have an interest in deciphering new roles for IDE and highlighting IDE as a potential target to treat the pathogenesis development in RP. However, I have concerns about some results or procedures shown in this manuscript.

Major comments:

  • Rationale for studding/exploring IDE expression in RP is not clearly stated in the manuscript.
  • What is the potential function associated with IDE that prevent or improve the pathogenesis of RP?
  • Although rescue experiments with sPIF show improvement of RP, the authors did not explore what is the underlying mechanism behind these effects.
  • In the Fig 2. B. the numbers that refer to the fluorescence intensity are difficult to see and to differentiate. For this reason, the authors should convert these numbers into scientific notation and increase its font size.
  • In 1. Results section, page 6 of 14, line 227; the authors mentioned that they used “The Human Protein Atlas” to explore the expression for IDE transcripts in the human retina. It would be great if the authors could add this graph to the Fig.1 or in Supplementary Figures.
  • The heading of sections 1 and 3.2 can be confusing because they are very similar. It would be better to change 3.2. for “Comparative IDE expression between WT vs RP retinas” for example.
  • In the Fig 3. A. while IDE RNA levels analysis are done in the three RP animal models, the IDE protein levels are only shown in P23H/P23H and rd10. Could please the authors justified why did they specifically use these models, and why rd1 was not included. Similarly, the authors should justified why they did only checked IDE activity in rd10 vs WT and not in the other two RP models.
  • In the same Fig 3., Western blots analysis of IDE expression should be shown in the same gel or blot. This mean that all samples must be round together, instead of showing them in separate independent panels. Duplicates of each samples would be also needed.
  • In 3. Treatment with sPIF ameliorated RP progression section, page 10 of 14, line 306, where the author results show that the treatment with sPIF improve ONL thickness, it should be previously explained that ONL thickness is a significant hallmark of the disease. In the same line, I suggest the authors to briefly explain some general notions of the potential pathological consequences of this disease on the retina (layers, etc), showing why is important the distribution of IDE as well.
  • Regarding the Figure 4. the authors stated page 8 line 265: “Furthermore, immunostaining of IDE showed a general decrease of IDE levels in dystrophic retinas”. However, this reviewer believes that the images showed in Figure 4 do not reflect what is stated in this sentence. Indeed, IDE staining observed for most of the images is diffuse, and it seems to be unspecific, except for some green dots showed in layer IS. Also, IDE staining in panels E-F, corresponding to GCL layers is hard to see, therefore it would be helpful to show the negative control assessed for these analyses (using same Ig in substitution of the primary antibody). Finally, the authors stated that the decrease of IDE levels in the layers could be the potential underlying cause of the pathology. However, if during RP shortens ONL, IS and OS segments, won´t be IDE reduction expected due to degeneration of these layers where IDE is normally expressed?

  • In 3. Treatment with sPIF ameliorated RP progressionsection, Fig 6., rescue IDE experiments should be test complement, not only with IP, but also should be assessed by IF and WB experiments (as shown previously in Fig.2 C and D). This is very important to demonstrate that the disease-modifying effect of sPIF throughout RP is due to a stimulation of IDE. In Fig 6. There is an evident improvement of photoreceptor preservation with sPIF treatment, but what is happening here with IDE? Is the sPIF beneficial effect due to change in IDE levels? In the Supplementary Fig 1. A. it is shown a significant increase in RNA levels, while there is not a significant change in IDE activity, but protein levels are not shown. Authors should present what happens with IDE protein levels. For this reason, IF, WB, or both experiments are indispensable to prove that the progression found with sPIF treatment is due to the rescue of IDE.
  • The heading of Fig 5. and Fig 6. should be ended like: ¨… on RP mice models.¨.

Minor comments:

  • There is some information in the manuscript that is missense:
    • The abbreviation “RT” for “room temperature” on page 3 of 14, line 127, should be changed because it can be confused with the acronym used for “reverse transcription” on page 3 of 14, line 114.
    • A reference is missing on page 2 of 14, line70, for justifying the role of sPIF as IDE activator.
    • Throughout the manuscript insulin degrading-enzyme can be found written in two different ways: IDE or Ide (example: page 5 of 14; line208 and line209). Authors must choose a single manner of naming this word. Then they should review the whole manuscript to confirm they have referred to it, in the same way, all the time.
    • In the same way, authors must be careful with “mice” or “mouse”. This is not mandatory, but it will improve the unified vision of the manuscript.
    • Similarly, “vehicle” and “saline” have been used interchangeably. Authors are recommended to unify this term. Since most of the time in the manuscript the authors have used “vehicle” in Fig 6. B and the term “saline” should be replaced with “vehicle” as it is noted in Fig 5, Fig 6. A. and S1.

  • Please check carefully that all abbreviations are indicated with () the first time that it appears in the text. Examples:
    • Inner segment (IS), page 2 of 14, line 67.
    • Ganglion cell layer (GCL), page 2 of 14, line 68.
    • Inner nuclear layer (INL), page 4 of 14, line153.
    • Outer nuclear layer (ONL), page 4 of 14, line 153.
    • Synthetic preimplantation factor (sPIF), page 2 of 14, line 70.
  • As it indicated for “rhodopsin” and “S-opsin” (on page 6 of 14, line 225), it would be nice if authors specify which is the difference of the cells stained with L/M- vs S- opsin (on page 10 of 14, line 311). As a suggestion from the referee, it would be a great idea to add a column in Table 2., clarifying the cell type that each antibody usually stains. In this way, readers who are not used to these stains will fully understand the immunofluorescence images.

  • To refer to a whole Figure is not necessary to indicate all the subsections (A, B…X ) that includes. For example, to mention Figure 6, specifying A and B after pointing out: Fig 6. is not required (as it appears on page 10 of 14, line 309). For this reason, on the same page, lines 307 and 313, it is over-understood that A and B are included in the reference just by mentioning Fig. 6.

Author Response

We thank the reviewers for the positive comments on our work, as well as the pertinent suggestions that have improved the precision of our manuscript. We would like to point out that we have tried to address all the comments raised by the reviewers.

All modifications and insertions in the manuscript are highlighted.

We hope the manuscript is now suitable for publication in Cells.

Reviewer 2

Comments and Suggestions for Authors

Review of the manuscript # cells-1649249 titled: Possible role of insulin-degrading enzyme in the physiopathology of retinitis pigmentosa.

In this manuscript, Sánchez-Cruz and colleagues explore the role of insulin degrading-enzyme (IDE) in the mammalian retina and its implication in retinitis pigmentosa (RP). The authors studied the expression and distribution of IDE in physiological and pathological conditions. For this reason, they used three RP mice models (rd1, rd10, P23H/P23H) that differ in the causative mutation. They found lower levels of IDE expression and protein in dystrophic retinas than in wild types (WT). Additionally, their results exhibit a decrease of IDE expression in the inner segment (IS) and the ganglion cell layer (GCL) of rd10 and P23H/P23H, whereas they previously showed that both areas present elevated IDE expression in WT. Then the authors analyzed whether IDE activation could reduce the pathological effects on the mammalian retina, using a synthetic preimplantation factor (sPIF), which has an activator effect over IDE expression. Interestingly, they found that sPIF treatment in rd10 mice improves the retinal response to light and promotes the preservation of photoreceptors.

Overall, the manuscript could have an interest in deciphering new roles for IDE and highlighting IDE as a potential target to treat the pathogenesis development in RP. However, I have concerns about some results or procedures shown in this manuscript.

Major comments:

  • Rationale for studding/exploring IDE expression in RP is not clearly stated in the manuscript.

As a follow up study to get more insight into the neuroprotective effect of proinsulin previously reported by our group (Fernández-Sánchez 2012; Isiegas 2016) we performed an exploratory experiment to investigated the mediators of such neuroprotective effect. We found that IDE protein levels were reduced in the rd10 retina, whereas proinsulin treatment significantly restored IDE protein levels. This prompted us to study the role of IDE in the physiopathology of the retina. We have included a sentence in the Introduction section in this respect (Ln 67-71), in addition to the previously referred sentence in the Discussion section (Ln 407-409).

  • What is the potential function associated with IDE that prevent or improve the pathogenesis of RP?

We currently do not know the precise function of IDE in the retina that plays a role in RP progression. However, building upon our knowledge accumulated along years we believe that IDE protective role is neither related to insulin clearance, since proinsulin and insulin are neuroprotective factors in the developing and diseased retina, nor to Aβ clearance since there is not Aβ accumulation in the RP retina. We think that IDE retinal functions may rather be related to its non-proteolytic functions. We have two favorite hypothesis.  On the one hand, photoreceptors are highly metabolic neurons that rely on their abundant mitochondrial content, clustered in the IS, to meet their high energetic and biosynthetic demands. The high IDE levels in the IS together with the existence of a mitochondrial IDE isoform may suggest that IDE plays a role in mitochondrial homeostasis. This is a tempting speculation that merits further investigations as we had mentioned in the Discussion section (Ln387-389). On the other hand, IDE protection could derive from its putative role as a scaffold protein. Photoreceptor OS are renewed daily and the IS contains all the machinery needed to the synthesis, ensemble and export of OS constituents to their final location. IDE could form part of this supramolecular machinery and thus contribute to photoreceptor structure maintenance as we had commented in the Discussion section (Ln396-398).

Although rescue experiments with sPIF show improvement of RP, the authors did not explore what is the underlying mechanism behind these effects. We share with the reviewer the interest in the mechanism behind the neuroprotective effect of sPIF in the context of RP. However, giving to the multiple potential effects of sPIF, this is a rather demanding task that excess the scope of this paper. Nevertheless, just for the reviewer information, in a preliminary study we have found that sPIF treatment decreases the expression of inflammatory markers in the rd10 retina. In future studies we intend to pursue this effect to better characterize the sPIF neuroprotective effect in the dystrophic retina.

  • In the Fig 2. B. thenumbers that refer to the fluorescence intensity are difficult to see and to differentiate. For this reason, the authors should convert these numbers into scientific notation and increase its font size. Thank you for the observation. We have change the notation and scale of the numbers in new Fig 2.D.  
  • In  Resultssection, page 6 of 14, line 227; the authors mentioned that they used “The Human Protein Atlas” to explore the expression for IDE transcripts in the human retina. It would be great if the authors could add this graph to the Fig.1 or in Supplementary Figures. We have included these data as a graph in Fig. 2E.
  • The heading of sections 1and 2 can be confusing because they are very similar. It would be better to change 3.2. for “Comparative IDE expression between WT vs RP retinas” for example. We have changed the heading of 3.2 as the reviewer suggested.
  • In the Fig 3. A.while IDE RNA levels analysis are done in the three RP animal models, the IDE protein levels are only shown in P23H/P23H and rd10. Could please the authors justified why did they specifically use these models, and why rd1 was not included. Similarly, the authors should justified why they did only checked IDE activity in rd10 vs WT and not in the other two RP models. All our animal experimental designs are guided by the “3R principle” as advised by national and international legislations in scientific procedures. To do each and every one of the experiments in all three animal models would have required to greatly increase the amount of animals. We started with the three RP models to address the mutation-independent downregulation of IDE in the context of RP. Then, we continued with the P23H/P23H and rd10 mouse models because they carry the most diverse mutations (different affected genes and distinct hereditary and disease progression patterns) while the rd1 and rd10 models harbor mutations in the same gene and have the same hereditary pattern as we described in M&M section. Finally, we performed the functional assays in the rd10 mice because this model is the less aggressive one of the three and allows for a wider therapeutic window.
  • In the same Fig 3.,Western blots analysis of IDE expression should be shown in the same gel or blot. This mean that all samples must be round together, instead of showing them in separate independent panels. Duplicates of each samples would be also needed.

All the samples to be compared were run in the same gel and blotted together. We have replaced the images and added a sentence in M&M for the record (Ln 186-189). In addition, as suggested by the reviewer, duplicates for each biological sample have been run and blotted and the mean of the duplicates for each biological replicate is now plotted.

  • In  Treatment with sPIF ameliorated RP progression section, page 10 of 14, line 306, where the author results show that the treatment with sPIF improve ONL thickness, it should be previously explained that ONL thickness is a significant hallmark of the disease.. In the same line, I suggest the authors to briefly explain some general notions of the potential pathological consequences of this disease on the retina (layers, etc), showing why is important the distribution of IDE as well. We have added a sentence in this regard (Ln 338-344).
  • Regarding the Figure 4. the authors stated page 8 line 265: “Furthermore, immunostaining of IDE showed a general decrease of IDE levels in dystrophic retinas”. However, this reviewer believes that the images showed in Figure 4 do not reflect what is stated in this sentence. Indeed, IDE staining observed for most of the images is diffuse, and it seems to be unspecific, except for some green dots showed in layer IS. Also, IDE staining in panels E-F, corresponding to GCL layers is hard to see, therefore it would be helpful to show the negative control assessed for these analyses (using same Ig in substitution of the primary antibody). We thank the reviewer for the observation. We have performed the requested negative control (new figure 2A) and we confirmed the observed specificity of IDE enrichment in the cone IS. However, the IDE labeling in the rest of the retinal layers, including the GCL, was not so conclusive. Therefore, we focused in the IDE expression in the cone IS until we are able to confirm the IDE expression in other retinal layers. We have eliminated the panels related to IDE staining in the GCL. Finally, the authors stated that the decrease of IDE levels in the layers could be the potential underlying cause of the pathology. However, if during RP shortens ONL, IS and OS segments, won´t be IDE reduction expected due to degeneration of these layers where IDE is normally expressed? RP involve a primary dysfunction and death of rod photoreceptors followed by secondary loss of cones (Ln 61-63; Ln 338-340). At the analyzed stages of the rd10 and P23H/P23H retinas cone survival is not affected yet and the decrease of the ONL thickness is due mainly to the loss of rods. Moreover, we had already analyzed the IDE immunofluorescence associated to individual cone inner segments (Ln 155-159). Therefore, the observed IDE reduction is not a reflection of the ONL decrease. However, since the molecular mechanism of phenomenon is totally unknown, we cannot rule out that, at least in part, the IDE reduction could be due to the shortening and/or disorganization of the outer and inner segments or, the other way around, that IDE decrease could contribute to the shortening and/or disorganization of the outer and inner segments. That’s why we stated “Taken together, these results demonstrate an association between RP and downregulation of retinal IDE, independent of the causative mutation”.
  • In  Treatment with sPIF ameliorated RP progression section, Fig 6., rescue IDE experiments should be test complement, not only with IP, but also should be assessed by IF and WB experiments (as shown previously in Fig.2 C and D). This is very important to demonstrate that the disease-modifying effect of sPIF throughout RP is due to a stimulation of IDE. In Fig 6. There is an evident improvement of photoreceptor preservation with sPIF treatment, but what is happening here with IDE? Is the sPIF beneficial effect due to change in IDE levels? In the Supplementary Fig 1. A. it is shown a significant increase in RNA levels, while there is not a significant change in IDE activity, but protein levels are not shown. Authors should present what happens with IDE protein levels. For this reason, IF, WB, or both experiments are indispensable to prove that the progression found with sPIF treatment is due to the rescue of IDE.

As the reviewer suggested, we have added a new panel in Figure S1 (S1C) in which showing the effect of sPIF treatment on IDE protein levels analyzed by immunostaining. sPIF-treated mice displayed higher levels of IDE in their cone IS than the vehicle-treated counterparts.

  • The heading of Fig 5.and Fig 6. should be ended like: ¨… on RP mice models.¨. We have changed the ending of the Fig. 5 and 6 headings to “…. on rd10 mice”

Minor comments:

  • There is some information in the manuscript that is missense:
    • The abbreviation “RT” for “room temperature”on page 3 of 14, line 127, should be changed because it can be confused with the acronym used for “reverse transcription” on page 3 of 14, line 114. Thank you for the observation. We have eliminated the RT abbreviation for room temperature
    • A reference is missing on page 2 of 14, line70, for justifying the role of sPIF as IDE activator.

The reference has been added.

  • Throughout the manuscript insulin degrading-enzyme can be found written in two different ways: IDE or Ide (example: page 5 of 14; line208 and line209). Authors must choose a single manner of naming this word. Then they should review the whole manuscript to confirm they have referred to it, in the same way, all the time.

We have followed the “Guidelines for Formatting Gene and Protein Names” that varies depending on the organism. For Mice and rats: Gene symbols are italicized, with only the first letter in upper-case (e.g., Gfap). Protein symbols are not italicized, and all letters are in upper-case (e.g., GFAP).

  • In the same way, authors must be careful with “mice”or “mouse”. This is not mandatory, but it will improve the unified vision of the manuscript.

Following the reviewer recommendation, we have changed mouse by mice whenever possible.

  • Similarly, “vehicle” and “saline” have been used interchangeably. Authors are recommended to unify this term. Since most of the time in the manuscript the authors have used “vehicle” in Fig 6.B and the term “saline” should be replaced with “vehicle” as it is noted in Fig 5, Fig 6. A. and S1. We have replaced the Word “saline” with “vehicle” as the reviewer suggested.

  • Please check carefully that all abbreviations are indicated with () the first time that it appears in the text. Examples:
    • Inner segment (IS), page 2 of 14, line 67.
    • Ganglion cell layer (GCL), page 2 of 14, line 68.
    • Inner nuclear layer (INL), page 4 of 14, line153.
    • Outer nuclear layer (ONL), page 4 of 14, line 153.
    • Synthetic preimplantation factor (sPIF), page 2 of 14, line 70.

We have reviewed the manuscript to define abbreviations when they first appear 

  • As it indicated for “rhodopsin” and “S-opsin” (on page 6 of 14, line 225), it would be nice if authors specify which is the difference of the cells stained with L/M- vs S- opsin (on page 10 of 14, line 311). As a suggestion from the referee, it would be a great idea to add a column in Table 2., clarifying the cell type that each antibody usually stains. In this way, readers who are not used to these stains will fully understand the immunofluorescence images. While all of the rods express rhodopsin in the mouse retina there are cones that express either S-opsin or M-opsin or both so analyzing both S-opsin and L/M-opsin we cover the whole cone population. This information has been added to the text (Ln 349-351)

We have added the information about cell type antibody specificity to Table 2. 

  • To refer to a whole Figure is not necessary to indicate all the subsections (A, B…X ) that includes. For example, to mention Figure 6, specifying A and B after pointing out:Fig 6. is not required (as it appears on page 10 of 14, line 309). For this reason, on the same page, lines 307 and 313, it is over-understood that A and B are included in the reference just by mentioning Fig. 6. We have made the changes requested by the reviewer.

Reviewer 3 Report

Retinitis pigmentosa is a group of retinal genetic dystrophies responsible for the majority of hereditary blindness cases.  Currently, retinitis pigmentosa is neither preventable nor curable.  Thus, the main research objectives of this study were to discover novel therapeutic targets for retinitis pigmentosa.  The authors showed that insulin degrading enzyme was widely expressed in the retina of wild type mice, with higher levels observed in photoreceptor inner segments, especially those of cones, and in the ganglion cell layer.  Gene expression and protein levels of insulin degrading enzyme were reduced in retinas of retinitis pigmentosa model with compare to age-matched wild type mice.  Furthermore, treatment with synthetic preimplantation factor, an activator of insulin degrading enzyme, resulted in better preservation of retinal structure and function in the retinitis pigmentosa mouse model.  The authors conclude that the present findings point to a novel role of insulin degrading enzyme in the retinal pathophysiology and beneficial effects of insulin degrading enzyme on retinal neurodegeneration will need to be corroborated in further studies using selective activators in vivo.  The manuscript is well-written and the methods sound.  I did not have any major concerns, only several minor issues listed below:

Page 2 lines 70–71, “Moreover, treatment with sPIF, an activator of IDE, results in better preservation of retinal structure and function in the rd10 mouse.”, Please define “sPIF” here.

Page 4 lines 142–144, “For the quantification of IDE intensity in the IS of cones we used the “freehand line” tool in Fiji software to select the whole length of the cone IS.”, Please define “IS” here.

Page 4 lines 149–151, “The intensity of IDE in the GCL was quantified measuring the mean intensity in 20 previously-defined regions of interest per mouse from 4 different images of the central retina.”, Please define “GCL” here.

Page 4 lines 152–154, “To assess preservation of the photoreceptor layer we compared the thickness of the ONL (which primarily contains photoreceptors) with that of the corresponding INL (which contains bipolar, horizontal, and amacrine neurons and Müller glial cell bodies).”, Please define “ONL” and “INL” here.

Author Response

We thank the reviewers for the positive comments on our work, as well as the pertinent suggestions that have improved the precision of our manuscript. We would like to point out that we have tried to address all the comments raised by the reviewers.

All modifications and insertions in the manuscript are highlighted.

We hope the manuscript is now suitable for publication in Cells.

Reviewer 3

Comments and Suggestions for Authors

Retinitis pigmentosa is a group of retinal genetic dystrophies responsible for the majority of hereditary blindness cases.  Currently, retinitis pigmentosa is neither preventable nor curable.  Thus, the main research objectives of this study were to discover novel therapeutic targets for retinitis pigmentosa.  The authors showed that insulin degrading enzyme was widely expressed in the retina of wild type mice, with higher levels observed in photoreceptor inner segments, especially those of cones, and in the ganglion cell layer.  Gene expression and protein levels of insulin degrading enzyme were reduced in retinas of retinitis pigmentosa model with compare to age-matched wild type mice.  Furthermore, treatment with synthetic preimplantation factor, an activator of insulin degrading enzyme, resulted in better preservation of retinal structure and function in the retinitis pigmentosa mouse model.  The authors conclude that the present findings point to a novel role of insulin degrading enzyme in the retinal pathophysiology and beneficial effects of insulin degrading enzyme on retinal neurodegeneration will need to be corroborated in further studies using selective activators in vivo.  The manuscript is well-written and the methods sound.  I did not have any major concerns, only several minor issues listed below:

Page 2 lines 70–71, “Moreover, treatment with sPIF, an activator of IDE, results in better preservation of retinal structure and function in the rd10 mouse.”, Please define “sPIF” here.

Done

Page 4 lines 142–144, “For the quantification of IDE intensity in the IS of cones we used the “freehand line” tool in Fiji software to select the whole length of the cone IS.”, Please define “IS” here.

We have introduced this abbreviation earlier in the new version (Page 2, Ln 76).

Page 4 lines 149–151, “The intensity of IDE in the GCL was quantified measuring the mean intensity in 20 previously-defined regions of interest per mouse from 4 different images of the central retina.”, Please define “GCL” here.

We have eliminated this piece of results from the manuscript (please see the response to reviewer 2)

Page 4 lines 152–154, “To assess preservation of the photoreceptor layer we compared the thickness of the ONL (which primarily contains photoreceptors) with that of the corresponding INL (which contains bipolar, horizontal, and amacrine neurons and Müller glial cell bodies).”, Please define “ONL” and “INL” here.

Done

Round 2

Reviewer 2 Report

We would like to congratulate the authors for their contribution in this study and also, to acknowledge their efforts to provide responses to all our suggestions. We hope this had helped to improve the quality of your work.